# Phytochemical Screening and Antibacterial Activity of Commercially Available Essential Oils Combinations with Conventional Antibiotics against Gram-Positive and Gram-Negative Bacteria

**DOI:** 10.3390/antibiotics13060478

**Published:** 2024-05-23

**Authors:** Răzvan Neagu, Violeta Popovici, Lucia-Elena Ionescu, Viorel Ordeanu, Andrei Biță, Diana Mihaela Popescu, Emma Adriana Ozon, Cerasela Elena Gîrd

**Affiliations:** 1Department of Pharmacognosy, Phytochemistry, and Phytotherapy, Faculty of Pharmacy, Carol Davila University of Medicine and Pharmacy, 6 Traian Vuia Street, 020956 Bucharest, Romania; razvan.neagu@drd.umfcd.ro (R.N.); cerasela.gird@umfcd.ro (C.E.G.); 2Regenerative Medicine Laboratory, “Cantacuzino” National Military Medical Institute for Research and Development, 103 Spl. Independentei, 050096 Bucharest, Romania; popescu.diana@cantacuzino.ro; 3Center for Mountain Economics, “Costin C. Kiriţescu” National Institute of Economic Research (INCE-CEMONT), Romanian Academy, 725700 Vatra-Dornei, Romania; 4Experimental Microbiology Laboratory, “Cantacuzino” National Military Medical Institute for Research and Development, 103 Spl. Independentei, 050096 Bucharest, Romania; 5Faculty of Pharmacy, “Titu Maiorescu” University, 16 Sincai, 040314 Bucharest, Romania; viorel.ordeanu@prof.utm.ro; 6Department of Pharmacognosy & Phytotherapy, Faculty of Pharmacy, University of Medicine and Pharmacy of Craiova, 200349 Craiova, Romania; andrei.bita@umfcv.ro; 7Department of Pharmaceutical Technology and Biopharmacy, Faculty of Pharmacy, Carol Davila University of Medicine and Pharmacy, 6 Traian Vuia Street, 020956 Bucharest, Romania; emma.budura@umfcd.ro

**Keywords:** essential oils, GC-MS, antibacterial activity, Gram-positive and Gram-negative bacteria, diffusimetric antibiogram, antibacterial drugs, combinations, interactions

## Abstract

The present study aims to evaluate the antibacterial activity of five commercially available essential oils (EOs), Lavender (LEO), Clove (CEO), Oregano (OEO), Eucalyptus (EEO), and Peppermint (PEO), against the most-known MDR Gram-positive and Gram-negative bacteria—*Staphylococcus aureus* (ATCC 25923), *Escherichia coli* (ATCC 25922), and *Pseudomonas aeruginosa* (ATCC 27853)—alone and in various combinations. Gas Chromatography–Mass Spectrometry (GC-MS) analysis established their complex compositions. Then, their antibacterial activity—expressed as the inhibition zone diameter (IZD) value (mm)—was investigated in vitro by the diffusimetric antibiogram method, using sterile cellulose discs with Ø 6 mm impregnated with 10 µL of sample and sterile borosilicate glass cylinders loaded with 100 µL; the minimum inhibitory concentration (MIC) value (µg/mL) for each EO was calculated from the IZD values (mm) measured after 24 h. The following EO combinations were evaluated: OEO+CEO, CEO+EEO, CEO+PEO, LEO+EEO, and EEO+PEO. Then, the influence of each dual combination on the activity of three conventional antibacterial drugs—Neomycin (NEO), Tetracycline (TET), and Bacitracin (BAC)—was investigated. The most active EOs against *S. aureus* and *E. coli* were LEO and OEO (IZD = 40 mm). They were followed by CEO and EEO (IZD = 20–27 mm); PEO exhibited the lowest antibacterial activity (IZD = 15–20 mm). EEO alone showed the highest inhibitory activity on *P. aeruginosa* (IZD = 25–35 mm). It was followed by CEO, LEO, and EEO (IZD = 7–11 mm), while PEO proved no antibacterial action against it (IZD = 0 mm). Only one synergic action was recorded (OEO+CEO against *P. aeruginosa*); EEO+PEO revealed partial synergism against *S. aureus* and CEO+PEO showed additive behavior against *E. coli*. Two triple associations with TET showed partial synergism against *E. coli*, and the other two (with NEO and TET) evidenced the same behavior against *S. aureus*; all contained EEO+PEO or CEO+PEO. Most combinations reported indifference. However, numerous cases involved antagonism between the constituents included in the double and triple combinations, and the EOs with the strongest antibacterial activities belonged to the highest antagonistic combinations. A consistent statistical analysis supported our results, showing that the EOs with moderate antibacterial activities could generate combinations with higher inhibitory effects based on synergistic or additive interactions.

## 1. Introduction

Currently, treating bacterial diseases is a challenge for modern medicine due to the expansion of antibiotic resistance. It is well known that numerous infections are induced by bacterial species present in many people without producing disease. Generally, these strains are non-pathogenic; however, these bacteria could become pathogenic when found in a place different from where they are commonly located or when their number significantly increases due to immunosuppression. According to the Federal Office of Public Health, the most common conditions are connected with the following bacterial species: *S. aureus*, *S. pneumoniae*, *E. coli*, *K. pneumoniae*, *A. baumani*, and *P. aeruginosa* [1]. All mentioned bacteria can quickly develop resistance to numerous conventional antibiotics [1]. Medical research focuses on finding and developing new inhibitory agents for multidrug-resistant (MDR) bacteria. Another potential strategy is to combine classical antibiotic drugs with plant-derived antimicrobials: various plant extracts, phytochemicals, and essential oils (EOs).

The EO chemical composition [2] includes phenolic compounds, terpenes, terpenoids [3], phenylpropanoids, and other aliphatic and aromatic constituents with strong lipophilic properties [4]. Their content varies depending on seasonal variations, climate, sub-species, and even the oil extraction method [5,6], which can lead to consequences for their antibacterial activity [7]. Investigating the EO antibacterial action is essential, considering the interactions between the main constituents and various mechanisms [8]. For example, OH phenolic from terpenoids is responsible for antibacterial activity and dual redox behavior. Thus, Eugenol, Linalool, and Linalyl-acetate block extracellular enzyme synthesis, inhibit histidine decarboxylase’s and ATPase’s enzymatic activity, and induce cell membrane depletion, losing ATP and K [9,10]. Other major constituents of various EOs—1,8-cineol and menthol derivatives (isomenthol and neomenthol)—exhibit significant antibacterial activity through bacterial surface charge alteration. They increase intracellular oxidative stress, inducing membrane disruption and bacterial damage [11,12]. Thymol induces extensive membrane destruction, inhibiting bacterial growth, decreasing intracellular ATP, cell membrane depolarization, and membrane damage [13]. Other EOs’ specific constituents, terpenes (p-Cymene, α-Pinene, γ-Terpinene, and Limonene), exhibit very low or no antibacterial activity; they can increase the antibacterial potential of other compounds. For example, p-cymene has a strong affinity for bacterial membranes, which changes their potential. This could affect the membrane permeability and decrease cellular motility [14].

Depending on each EO phytochemical profile, the antibacterial effects could be due to cellular membrane disruption through protein damage, cell content leakage, motive proton force depletion, and cytoplasm coagulation [15,16,17]. The initial negative impact of EOs on the bacterial cell membrane influences the spectrum of bacterial metabolic processes affected by the responsible EO phytocompounds (ATPase, lipase, and coagulase activity inhibition, leakage of cell ions, citrate metabolic pathway disruption, efflux pump blockage, and fluidization of membrane lipids) [18,19,20,21]. Moreover, they can inhibit bacterial toxins production (α-toxin, enterotoxins A and B, and listeriolisin), diminishing their virulence [22,23]. Various scientists have combined different EOs to identify their synergistic associations [24] for therapeutic or food preservative purposes [20]. Moreover, based on differences in antibacterial mechanisms, EOs were coupled with conventional antibiotics [16], because most antibacterial drugs act by a cell wall and protein synthesis inhibition [25]. These combinations were tested against MDR bacterial strains to determine their usefulness. The most studied synergies imply EOs of *Melaleuca alternifolia*, *Coriandrum sativum*, *Lippia sidoides*, *Thymus maroccanus*, *Cinnamomum zeylanicum*, *Syzygium aromaticum*, *Mentha piperita*, *Origanum vulgare*, and *Rosmarinus officinalis* combined with antibiotic drugs from various classes (beta-lactams, quinolones, aminoglycozides, chloramphenicol, tetracyclin, and polypeptides) against MDR Gram-positive and Gram-negative bacteria [26].

In this context, the present study aims to investigate the antibacterial activity and potential interactions of five commercially available EOs—Lavender (LEO), Clove (CEO), Oregano (OEO), Eucalyptus (EEO), and Peppermint (PEO)—in binary combinations and triple associations with antibiotic drugs—Tetracycline (TET), Neomycin (NEO), and Bacitracin (BAC) against *S. aureus*, *E coli*, and *P. aeruginosa*. A comparative GC-MS analysis of autochthonous EOs and standard ones, rarely used antibiotic drugs, and selected EO combinations investigated through two different techniques of diffussimetric antibiogram, by using paper discs (disc diffusion method—DDM) and borosilicate glass cylinders (cylinder technique—CT), represents the novelty of our research.

## 2. Results

### 2.1. Gas Chromatography–Mass Spectrometry Analysis

The GC-MS chromatograms of the tested EOs are displayed in Figure 1. The main quantified constituents, expressed as percentages (%), are registered in Table 1. For each EO, both its sample and standard compositions are displayed. The GC-MS chromatograms of the EOs used as standards are in the Appendix A. Moreover, all constituents identified and quantified in all EOs are in Appendix A.

The main constituents of each EO are as follows: Eugenol (86.22%), Caryophyllene (6.87%), and Eugenol acetate (5.75%) in CEO; Eucalyptol (83.74%), Limonene (5.45%), o-Cymene (4.13%), α-Pinene (2.81%), and γ-Terpinene (2.23%) in EEO; Linalool (52.93%), Linalyl acetate (32.31%), α-Pinene (3.16%), Nerol (2.11%), Camphor (1.34%), and Limonene (1.12%) in LEO; Neoisomenthol (55.09%), Isomenthone (26.52%), Eucalyptol (5.04%), Menthofuran (2.46%), Limonene (2.08%), and Caryophyllene (1.61%) in PEO; and p-Thymol (72.08%), o-Cymene (16.26%), γ-Terpinene (2.98%), α-Pinene (2.39%), Caryophyllene (1.45%), Linalool (1.40%), and β-Pinene (1.39%) in OEO (Figure 1 and Table 1).

Eucalyptol, γ-Terpinene, Linalool, Terpinen 4-ol, Limonene, o-Cymene, α-Pinene, α-Myrcene, β-Pinene, and Camphene are common constituents in four EOs: EEO, LEO, PEO, and OEO. Caryophylene is found in CEO, PEO, and OEO. The specific compounds with the highest contents are Eugenol and Eugenyl acetate in CEO, Isomenthone, Neoisomenthone, Menthofuran, and Eucalyptol in PEO, p-Thymol, γ-Terpinene, and o-Cymene in OEO, Linalool, Linalyl-acetate, α-Pinene, and Nerol in LEO, and Eucalyptol, γ-Terpinene, Limonene, o-Cymene, and α-Pinene in EEO (Figure 1, Appendix A.

In some EO samples, the main constituents’ contents were higher than those in the standards (Eucalyptol in EEOs—83.7478 > 81.9587%; Linalool in LEOs—52.9348 > 35.8533%; Neoisomenthol in PEOs—55.0951 > 50.8339%; and Eugenol-acetate in CEOs—5.7515 > 1.5998%). Contrariwise, in other EO samples, they had lower contents than those in the standards (Eugenol and Caryophyllene in CEOs—86.2272 < 87.0884%; 6.8733 < 9.8462%; p-Thymol and γ-Terpinene in OEOs—72.0862 < 77.9238%; 2.9827 < 5.1776%, and Linalyl-acetate, o-Cymene, and α-Pinen in LEOs—32.3146 < 36.2963%, 0.2470 < 3.0292%, and 3.1612 < 5.6669%).

### 2.2. Antibacterial Activity on Gram-Positive and Gram-Negative Bacteria

Five binary EO combinations were performed based on the EO phytochemical profiles and our previous screening [27]. CEO’s constituents differ in the highest measure from all other EOs tested (Figure 1, Appendix A). Therefore, CEO, EEO, and PEO were combined in three pairs (CEO+EEO, CEO+PEO, and EEO+PEO); two other EOs were added in only two combinations: OEO+CEO and LEO+EEO. Each binary combination was combined with three conventional antibiotics: TET, NEO, and BAC, resulting in 15 triple combinations.

The selected antibiotic drugs belong to three different classes. Tetracycline is a broad-spectrum polypeptide antibiotic that exerts a bacteriostatic effect by reversibly binding bacterial 30S ribosomal subunit and blocking protein synthesis [28]. Neomycin is a broad-spectrum aminoglycoside [29] active against Gram-positive and Gram-negative bacteria by linking cellular ribosomes and inhibiting protein synthesis [30]. Bacitracin is a polypeptide antibacterial drug that acts against Gram-positive bacteria by inhibiting cell wall synthesis [30].

For an accurate evaluation of plant-derived products’ antibacterial potential, Vorobets et al. recommend using at least two different techniques of diffusimetric antibiogram [31]. Therefore, we selected the disk diffusion method (DDM) [32] and cylinder technique (CT) [33]. The data obtained are presented synthetically in Table 2, Table 3 and Table 4.

The scale of measurement was as follows: a powerful inhibitory effect at IZD ≥ 35 mm, strong inhibitory effects at 35 > IZD ≥ 25 mm, a moderate inhibitory effect at 25 > IZD ≥15 mm, a mild inhibitory effect when 15 > IZD ≥ 10 mm, and a very low inhibitory effect at IZD < 10 mm [34]. Several similarities were observed between the IZD (inhibition zone diameter) was measured through both methods. However, these inconsistencies were due to the different diffusion levels of the samples’ constituents in the culture medium, especially when they were tested in a large volume (100 µL).

#### 2.2.1. Antibacterial Activity against *S. aureus*

*S. aureus* represents a part of skin flora in approximately 30% of humans without causing disease. However, *S. aureus* can induce skin, bone, and bloodstream infections in immunocompromised patients; it is the most frequent bacteria implied in post-surgical infections. [1]. The data from Table 2 show that LEO and OEO displayed a powerful inhibitory effect on *S. aureus* (IZD = 40 mm). Similar results can also be found in other studies [35,36]. PEO and EEO had the lowest effects measured by DDM; the same IZD range (15–24 mm) belongs to the values reported by other researchers [37,38]. However, their combination (EEO+PEO) tested by DDM was partially synergistic (FICI = 0.8), with a substantial inhibitory effect (IZD = 35 mm), significantly higher than each EO (Table 2). The same binary EWO combination, tested by CT, showed indifference (FICI = 1.4).

The corresponding *p*-values for each technique are displayed in
Appendix A.

All EOs and combinations exhibited inhibitory activity against *S. aureus*. The anti-staphylococcal effect of antibiotic drugs increased in the order of NEO < BAC < TET; combined with EEO+PEO, NEO and TET generated partially synergistic triple combinations when cylinder technique was used (Table 2).

**Table 2 antibiotics-13-00478-t002:** The antibacterial effects of EOs combinations and classical antibiotics on *S. aureus*, evaluated by both techniques of diffusimetric method, registered as a mean of 3 repetitions: IZD (mm), MIC (µg/mL), and FICI (for double and triple combinations).

Technique	Cellulose Disc Technique (DDM)					Cylinder Technique (CT)						
Sample	*S. aureus*											
	IZD (mm)		MIC (µg/mL)	FICI			IZD (mm)		MIC (µg/mL)	FICI		
	Range	Value		Range		Value	Range	Value		Range		Value
LEO	≥35	40	2.1	-		-	≥35	40	21.2	-		-
CEO	25–34	27	4.7	-		-	25–34	25	54.3	-		-
OEO	≥35	40	2.1	-		-	≥35	40	21.2	-		-
EEO	15–24	20	8.5	-		-	25–34	25	54.3	-		-
PEO	25–34	25	5.4	-		-	15–24	15	150.9	-		-
OEO+CEO	25–34	30	3.8	>1	≤4	2.60	15–24	15	150.9	>4	≤10	9.90
CEO+EEO	15–24	15	15.1	>4	≤10	5.00	25–34	27.5	44.9	>1	≤4	1.60
CEO+PEO	15–24	15	15.1	>4	≤10	6.00	15–24	20	84.9	>1	≤4	2.20
LEO+EEO	≥35	40	2.1	>1	≤4	1.20	25–34	30	37.7	>1	≤4	2.50
EEO+PEO	≥35	35	2.8	>0.5	<1	0.80	25–34	25	54.3	>1	≤4	1.40
NEO	<10	7	86.7	-		-	15–24	20	106.2	-		-
OEO+CEO+NEO	25–34	25	5.7	>1	≤4	3.98	15–24	20	89.2	>4	≤10	6.67
CEO+EEO+NEO	25–34	25	5.7	>1	≤4	1.94	≥35	37.5	25.4	>1	≤4	1.15
CEO+PEO+NEO	10–14	12.5	22.8	>4	≤10	9.33	15–24	22	73.7	>1	≤4	2.52
LEO+EEO+NEO	15–24	23.5	6.5	>1	≤4	3.83	25–34	25	57.1	>4	≤10	4.27
EEO+PEO+NEO	25–34	26	5.3	>1	≤4	1.66	15–24	15	15.9	>0.5	<1	0.53
TET	25–34	25	6.8	-		-	25–34	29	50.5	-		-
OEO+CEO+TET	25–34	25	5.7	>4	≤10	4.87	≥35	40	21.2	>1	≤4	1.80
CEO+EEO+TET	15–24	22	7.4	>1	≤4	3.52	15–24	15	150.9	>4	≤10	8.52
CEO+PEO+TET	25–34	25	5.7	>1	≤4	3.09	≥35	45	16.8	>0.5	<1	0.74
LEO+EEO+TET	25–34	26	5.3	>4	≤10	4.83	25–34	30	37.7	>1	≤4	3.20
EEO+PEO+TET	25–34	26	5.3	>1	≤4	2.37	10–14	10	339.7	>15	≤20	15.22
BAC	15–24	17	14.7	-		-	25–34	25	67.9	-		-
OEO+CEO+BAC	25–34	30	4.0	>1	≤4	3.02	25–34	30	47.2	>1	≤4	3.77
CEO+EEO+BAC	25–34	32.5	3.4	>1	≤4	1.35	0	0	-	-		-
CEO+PEO+BAC	15–24	17.5	11.6	>4	≤10	5.38	0	0	-	-		-
LEO+EEO+BAC	25–34	25	5.7	>1	≤4	3.76	25–34	27	48.9	>4	≤10	5.32
EEO+PEO+BAC	25–34	32.5	3.4	>1	≤4	1.25	15–24	20	89.2	>1	≤4	3.54

LEO—Lavender Essential Oil; CEO—Clove Essential Oil; OEO—Oregano Essential Oil; EEO—Eucalyptus Essential Oil; PEO—Peppermint Essential Oil; NEO—Neomicin; TET—Tetracyclin; and BAC—Bacitracin (https://microbiologie-clinique.com/antibiotic-family-abbreviation.html, accessed on 5 February 2024). IZD—inhibition zone diameter (mm); the scale of measurement was as follows: very strong inhibitory effect at IZD ≥ 35 mm, strong inhibitory effects at 35 > IZD ≥ 25, moderate inhibitory effect at 25 > IZD ≥ 15 mm, mild inhibitory effect when 15 > IZD ≥ 10 mm, and very low inhibitory effect at IZD < 10 mm. MIC—minimum inhibitory concentration (µg/mL), FICI—fractional inhibitory concentration index. If IZD = 0 mm, the substance has no effect; there is no MIC. FICI ≤ 0.5 indicates synergism (S). 0.5 < FICI < 1 means partial synergism (PS); FICI = 1 indicates additive effects (Add.); 1 < FICI ≤ 4—indifference (I); FICI > 4 is antagonism (Ant.) [39]; 4 < FICI ≤ 10—low antagonism; 10 < FICI ≤ 15—moderate antagonism; 15 < FICI ≤ 20—strong antagonism; FICI > 20—very strong antagonism.

#### 2.2.2. Antibacterial Activity against *E. coli*

*E. coli* lives in the human gut, commonly without inducing disease. It can cause infections when it penetrates the urinary tract [1]; in human household contacts, dogs can be reservoirs of *E. coli* strains responsible for urinary tract infections [40]. Moreover, in immunosuppressed patients, *E. coli* can produce severe infections: otitis [41] and meningitis (in neonates and older adults [42]).

The data from Table 3 shows that OEO and LEO exhibited the highest antibacterial effect on *E. coli*. Conversely, PEO and BAC had no inhibitory activity against it, and CEO, EEO, TET, and NEO displayed moderate ones. In this context, using the DDM technique, CEO+PEO revealed additive antibacterial activity on *E coli*, and triple combinations with TET (CEO+PEO+TET and EEO+PEO+TET) acted partially synergistically. The corresponding *p*-values for each technique are displayed in
Appendix A.

**Table 3 antibiotics-13-00478-t003:** The antibacterial effect of essential oils (EOs) combinations and classical antibiotics on *E. coli* evaluated by
both techniques of diffusimetric method, registered as a mean of 3 repetitions:
IZD (mm), MIC (µg/mL), and FICI (for double and triple combinations).

Technique	Cellulose Disc Technique						Cylinder Technique					
Sample	*E. coli*											
	IZD (mm)		MIC (µg/mL)	FICI			IZD (mm)		MIC (µg/mL)	FICI		
	Range	Value		Range		Value	Range	Value		Range		Value
LEO	≥35	40	2.1	-		-	≥35	40	21.2	-		-
CEO	15–24	22	7.0	-		-	15–24	15	150.9	-		-
OEO	≥35	45	1.7	-		-	≥35	45	16.8	-		-
EEO	25–34	25	5.4	-		-	25–34	30	37.7	-		-
PEO	0	0	-	-		-	10–14	10	339.7	-		-
OEO+CEO	25–34	25	5.4	>1	≤4	4.00	15–24	15	150.9	>4	≤10	10.00
CEO+EEO	15–24	15	15.1	>4	≤10	4.90	10–14	10	339.7	>10	≤15	11.30
CEO+PEO	15–24	22	7.0	=1		1.00	0	0	-	-		-
LEO+EEO	10–14	12	23.6	>15	≤20	15.60	15–24	15	150.9	>10	≤15	11.10
EEO+PEO	10–14	13	20.1	>1	≤4	3.70	10–14	12	235.9	>4	≤10	7.00
NEO	15–24	15	18.9	-		-	25–34	25	67.9	-		-
OEO+CEO+NEO	25–34	30	4.0	>1	≤4	3.70	0	0	-	-		-
CEO+EEO+NEO	15–24	15	15.9	>4	≤10	6.05	25–34	27	48.9	>1	≤4	2.33
CEO+PEO+NEO	25–34	25	5.7	>1	≤4	1.11	15–24	20	89.2	>1	≤4	2.16
LEO+EEO+NEO	15–24	15	15.9	>10	≤15	11.35	25–34	25	57.1	>4	≤10	5.04
EEO+PEO+NEO	15–24	15	15.9	>1	≤4	3.78	15–24	20	89.2	>1	≤4	3.93
TET	10–14	14	21.7	-		-	25–34	30	47.2	-		-
OEO+CEO+TET	25–34	25	5.7	>4	≤10	4.42	25–34	25	57.1	>4	≤10	4.96
CEO+EEO+TET	25–34	26	5.3	>1	≤4	1.97	25–34	28	45.5	>1	≤4	2.46
CEO+PEO+TET	25–34	27	4.9	>0.5	<1	0.92	25–34	26	52.8	>1	≤4	1.59
LEO+EEO+TET	25–34	26	5.3	>1	≤4	3.74	25–34	27.5	47.2	>4	≤10	4.47
EEO+PEO+TET	25–34	30	4.0	>0.5	<1	0.92	25–34	30	39.6	>1	≤4	1.99
BAC	0	0	-	-		-	0	0	-	-		-
OEO+CEO+BAC	25–34	25	5.7	>4	≤10	4.16	15–24	15	158.5	>10	≤15	10.48
CEO+EEO+BAC	10–14	13	21.1	>4	≤10	6.91	<10	8	557.4	>15 20	≤20	18.47
CEO+PEO+BAC	15–24	20	8.9	>1	≤4	1.27	0	0	-	-		-
LEO+EEO+BAC	10–14	10	35.7	>20		23.61	10–14	10	356.7	>20		26.28
EEO+PEO+BAC	<10	9	44.0	>4	≤10	8.14	10–14	10	356.7	>10	≤15	10.51

LEO—Lavender Essential Oil; CEO—Clove Essential Oil; OEO—Oregano Essential Oil; EEO—Eucalyptus Essential Oil; PEO—Peppermint Essential Oil; NEO—Neomycin; TET—Tetracycline; BAC—Bacitracin; IZD—inhibition zone diameter (mm); the scale of measurement was as follows: very strong inhibitory effect at IZD ≥ 35 mm, strong inhibitory effects at 35 > IZD ≥ 25, moderate inhibitory effect at 25 > IZD ≥ 15 mm, mild inhibitory effect when 15 >IZD ≥ 10 mm, and very low inhibitory effect at IZD < 10 mm. MIC—minimum inhibitory concentration (µg/mL), FICI—Fractional inhibitory concentration; FICI ≤ 0.5 indicates synergism (S). 0.5 < FICI < 1 means partial synergism (PS); FICI = 1 indicates additive effects (Add.); 1 < FICI ≤ 4—indifference (I); FICI > 4 is antagonism (Ant.) [39]; 4 < FICI ≤ 10—low antagonism; 10 < FICI ≤ 15—moderate antagonism; 15 < FICI ≤ 20—strong antagonism; FICI > 20—very strong antagonism.

#### 2.2.3. Antibacterial Activity against *P. aeruginosa*

*P. aeruginosa* is most known as a nosocomial pathogen, mainly determining severe infections within hospitals and care homes. Its resistance to carbapenems and polymyxins is continuously increasing. Surgical and medical devices (eq., endoscopes) potentially spread this MDR bacteria, leading to nosocomial outcomes [43].

Table 4 indicates that PEO and BAC had no inhibitory effects on *P. aeruginosa*; most EO triple combinations with BAC acted similarly through DDM. Moreover, all the other EOs evidenced very low/low/moderate antibacterial activity on *P. aeruginosa*. One binary combination was synergistic—OEO+CEO, as evaluated by the cylinder technique. The corresponding *p*-values for each technique are displayed in Appendix A.

Table 4 also shows the lowest IZD values from all three tables and the highest MIC ones, indicating the modest inhibitory activity of all EOs alone or combined.

**Table 4 antibiotics-13-00478-t004:** The antibacterial activity of EO combinations and classical antibiotics against *P. aeruginosa*, evaluated by both techniques of diffusimetric method, registered as a mean of 3 repetitions: IZD (mm), MIC (µg/mL) and FICI (for double and triple combinations).

Technique	Cellulose Disc Technique						Cylinder Technique					
Sample	*P. aeruginosa*											
	IZD (mm)		MIC (µg/mL)	FICI			IZD (mm)		MIC (µg/mL)	FICI		
	Range	Value		Range		Value	Range	Value		Range		Value
LEO	10–14	11	28.1	-		-	<10	7	693.2	-		-
CEO	10–14	10	33.9	-		-	<10	8	530.9	-		-
OEO	10–14	11	28.1	-		-	10–14	10	339.7	-		-
EEO	25–34	25	5.4	-		-	≥35	35	27.7	-		-
PEO	0	0	-	-		-	0	0	-	-		-
OEO+CEO	<10	7.5	60.6	>1	≤4	3.90	25–34	30	37.7	≤0.5		0.20
CEO+EEO	<10	7.5	60.6	>10	≤15	13.00	≥35	35	27.7	>1	≤4	1.10
CEO+PEO	<10	7.5	60.6	>1	≤4	1.80	0	0	-	-		-
LEO+EEO	<10	8	53.0	>10	≤15	11.70	15–24	16	132.7	>4	≤10	5.00
EEO+PEO	<10	8	53.0	>4	≤10	9.80	15–24	17	117.5	>4	≤10	4.20
NEO	10–14	10	42.4	-		-	15–24	20	106.2	-		-
OEO+CEO+NEO	0	0	-	-		-	15–24	22	73.7	>1	≤4	1.16
CEO+EEO+NEO	10–14	12.5	22.8	>4	≤10	5.56	15–24	15	158.5	>4	≤10	7.50
CEO+PEO+NEO	15–24	15	15.9	>1	≤4	1.29	0	0	-	-		-
LEO+EEO+NEO	<10	9	44.0	>10	≤15	10.73	10–14	14	182.0	>1	≤4	2.05
EEO+PEO+NEO	15–24	17	12.3	>1	≤4	2.56	15–24	21	80.9	>1	≤4	3.68
TET	15–24	15	18.9	-		-	25–34	25	67.9	-		-
OEO+CEO+TET	0	0	-	-		-	<10	7	727.9	>10	≤15	14.23
CEO+EEO+TET	15–24	18	11.0	>1	≤4	2.93	15–24	23	67.4	>1	≤4	3.54
CEO+PEO+TET	10–14	14	18.2	>1	≤4	1.49	15–24	15	158.5	>1	≤4	2.62
LEO+EEO+TET	10–14	14	18.2	>4	≤10	4.97	15–24	15	158.5	>4	≤10	8.27
EEO+PEO+TET	15–24	15	15.9	>1	≤4	3.78	15–24	22	73.7	>1	≤4	3.74
BAC	0	0	-	-		-	0	0	-	-		-
OEO+CEO+BAC	0	0	-	-		-	0	0	-	-		-
CEO+EEO+BAC	0	0	-	-		-	15–24	22	73.7	>1	≤4	2.79
CEO+PEO+BAC	0	0	-	-		-	0	0	-	-		-
LEO+EEO+BAC	0	0	-	-		-	15–24	21	80.9	>1	≤4	3.03
EEO+PEO+BAC	<10	6	99.1	>15	≤20	18.35	15–24	20	89.2	>1	≤4	3.22

LEO—Lavender Essential Oil; CEO—Clove Essential Oil; OEO—Oregano Essential Oil; EEO—Eucalyptus Essential Oil; PEO—Peppermint Essential Oil; NEO—Neomycin; TET—Tetracyclin; and BAC—Bacitracin. IZD—inhibition zone diameter (mm); the scale of measurement was as follows: very strong inhibitory effect at IZD ≥ 35 mm, strong inhibitory effects at 35 > IZD ≥ 25, moderate inhibitory effect at 25 > IZD ≥ 15 mm, mild inhibitory effect when 15 > IZD ≥ 10 mm, and very low inhibitory effect at IZD < 10 mm. MIC—minimum inhibitory concentration (µg/mL), FICI—fractional inhibitory concentration. If IZD = 0 mm, the substance has no effect and no MIC; FICI ≤ 0.5—synergism (S). 0.5 < FICI < 1—partial synergism (PS); FICI = 1—additive effects (Add.); 1 < FICI ≤ 4—indifference (I); FICI > 4—antagonism (Ant.) [39]; 4 < FICI ≤ 10—low antagonism; 10 < FICI ≤ 15—moderate antagonism; 15 < FICI ≤ 20—strong antagonism; FICI > 20—very strong antagonism.

The data from Table 3 and Table 4 show that the IZD sizes recorded on the Gram-negative bacteria (especially on *P. aeruginosa*) through the cylinder technique are higher than those measured by DDM. Moreover, Vorobets et al. noted that, in their study, the cylinder technique was the most sensitive, recording the highest IZD values [31].

Table 1, Table 2 and Table 3 report a few EO binary combinations (eq. CEO+EEO) with an IZD lower than that of each EO alone. These results suggest similar inhibitory mechanisms, with some compounds’ competition for various binding sites.

Different data regarding antibacterial activity expressed as IZD (mm) for EOs, conventional antibiotics, and double and triple combinations are displayed in Figure 2 and the corresponding *p*-values are shown in Appendix A. Thus, we can see that double EO combinations lead to an increasing antibacterial activity against *P. aeruginosa* in the cylinder technique (Figure 2). Generally, using the cylinder technique, NEO reduces the EO inhibitory activity on the bacteria tested (Figure 2, except on *P. aeruginosa*. On the other hand, triple combinations with TET have a higher antibacterial activity on all tested strains compared to the double EO combinations. Contrariwise, BAC mixed with double EO combinations had diminished IZD values (Figure 2).

Despite the different volumes of the samples used in both techniques (10 µ in DDM and 100 µ in CT), the IZD ranges were similar. Only the calculated MIC values recorded considerable differences, because the calculation formula considered the size of the sample’s volume. The antibacterial potential was negatively correlated with MIC value. The DDM technique had the lowest MIC values and could be considered as better than CT.

The results of both techniques (DDM and CT, 1 and 2) of the diffusimetric method were compared using a Bland–Altman analysis. The results are displayed in the MS text, Figure 3, and Appendix A. Similar results (*p* < 0.05) were obtained by both techniques against *S. aureus* (*p* = 0.566) and *E. coli* (*p* = 0.460). Significant differences between DDM and CT were recorded in terms of antibacterial activity against *P. aeruginosa* (*p* = 0.002 < 0.05), according to Figure 3E. The triple combination EOs+TET were responsible for this (*p* = 0.036 < 0.05).

In Figure 3B,D,F are evidenced the differences between both techniques.

The main constituents of the tested EOs are mono- and sesquiterpene hydrocarbons and their oxygenated derivatives; they are volatile compounds sensitive to heating, air, and humidity. Therefore, the antibacterial activity of the EOs could be reduced during both technical processes by losses due to evaporation [44,45,46,47] at 37 °C. This process occurs more in DDM, because small sample volumes (10 µL) are incorporated in filter paper discs placed on a culture medium. Therefore, the EO bioactive constituents only partially diffuse in culture media, inhibiting bacterial strain growth. In CT, 10× higher volumes (100 µL) progressively diffuse in the culture media, thus diminishing the volatile compounds’ evaporation and increasing their biodisponibility.

On the other hand, some EO constituents absorbed in agar layers could suffer alteration processes during the incubation period of 24 h. Inouye et al. reported limonene and α-pinene oxidation [48]; in the present study, both constituents were found in four EOs tested (EEO, OEO, LEO, and PEO).

All these aspects could explain the different IZD sizes obtained through both techniques; depending on each EO phytochemical profile, one technique or the other could be more suitable, and the results more relevant and similar to those of previously published research. Thus, comparing the IZD values obtained through both methods, the most appropriate ones could be considered for the MIC and FICI calculation and interpretation.

Several studies reported that CEO is efficient against *S. aureus* and *E. coli* [49], OEO is active on all three bacteria, and PEO shows moderate or low activity on *S. aureus* and *P. aeruginosa* [50,51]. LEO and EEO exhibited inhibitory activity against *S. aureus* and *E. coli* [52,53]. Another study revealed the antibacterial activity of EEO emulsion against *E. coli* and *P. aeruginosa* [54].

The calculation of MIC values from the IZD ones led to evaluating the interaction between EOs in binary combinations and EOs and conventional antibiotics in triple combinations, possibly by analyzing each FIC index (Figure 4).

Figure 4 shows that, in a binary combination, OEO and CEO synergistically act against *P. aeruginosa* (PaFICI2 = 0.2); this result was assessed using the cylinder technique. Partial synergism was revealed by another EO pair (EEO+PEO) against *S. aureus* (SaFICI1 = 0.8) using DDM. EEO+PEO in triple combinations (with NEO and TET) evidenced partial synergism against *S. aureus* (SaFICI2 = 0.53) and *E. coli* (EcFICI1 = 0.92).

Previous studies reported that a 1, 8-Cineol+Linalool combination creates more resistance in *E. coli*, compared to applying pure Linalool [55], while Carvacrol and Thymol are synergistic [56]. Synergistic effects were revealed for Eugenol, Carvacrol, Thymol, p-Cymene, and γ-Terpinene [57] on inhibiting drug resistance and the biofilm formation of oral bacteria (*S. aureus*). The Thymol+Menthol combination was reported as being antagonistic against *E. coli* and *S. aureus*, while Geraniol+Menthol was synergistic against *S. aureus* and indifferent against *E. coli* [49]. Moreover, various authors revealed synergy between Eugenol with Chloramphenicol and Norfloxacin with Oxacillin against *E. coli* and *P. aeruginosa*, Menthol with Oxytetracycline against *E coli*, Thymol with Norfloxacin and Bacitracin against *S. aureus*, and Novobiocin with Penicillin against *E. coli* [26].

On the other hand, PEO was previously tested on *S. aureus* in association with Ciprofloxacin; against *E. coli*, PEO formed synergistic combinations with Ampicilin, Erytromicin, Oxytetracycline, and Gentamycin [44,45]. In our study, a binary EO combination (CEO+PEO) recorded additive effects against *E. coli* (EcFICI1 = 1), and another triple combination (CEO+PEO+TET) showed partial synergism against *S. aureus* and *E. coli* (SaFICI2 = 0.74 and EcFICI1 = 0.92). Previous studies revealed synergistic effects of CEO combined with Ampicillin and Gentamycin against another *Staphylococcus* sp., *S. epidermidis* [46].

Moreover, CEO+PEO+TET and EEO+PEO+NEO did not show antagonism (FICI < 4). Most combinations displayed indifference and/or low antagonism. There were no recorded binary or triple combinations with an exclusive antagonism against all bacteria tested (Figure 3). However, LEO+EEO+BAC revealed powerful antagonism against *E. coli* (FICI > 20) and LEO+EEO+NEO moderate/low effects; the binary EOs combination (LEO+EEO) showed antagonism on both Gram-negative bacteria (strong/moderate against *E. coli* and moderate/low against *P. aeruginosa*). In contrast, the third triple combination (LEO+EEO+TET) recorded low antagonism only against *P. aeruginosa*. Other triple combinations with BAC, CEO+EEO+BAC, EEO+PEO+BAC, and OEO+CEO+BAC, showed antagonistic interactions against *E coli* (low/strong or low/moderate, Figure 4), while CEO+EEO and OEO+CEO acted similarly, and EEO+PEO and CEO+EEO+NEO registered low agonism against *P. aeruginosa*. CEO+EEO+TET and EEO+PEO+TET did not record an agonist behavior against both Gram-negative bacteria, while CEO+PEO+TET and EEO+PEO+NEO showed only partial synergism and indifference against all bacteria by both methods (Figure 4).

### 2.3. Data Analysis

An extensive data analysis was performed to support our results.

First, the correlations between chemical composition and antibacterial activity (expressed as IZD) were analyzed through Principal Component Analysis (PCA) and are illustrated in Figure 5A–C. All correlation biplots in Figure 5 show the place of each EO associated with its phytochemical profile and antibacterial activity.

Figure 5A is designed for the first group of three EOs (CEO, PEO, and EEO) and is consequently associated with three binary combinations. Figure 5B shows the same correlations in the second EO group (CEO, EEO, OEO, and LEO) associated with the other two binary combinations. Figure 5C presents an overview of the correlation of all five EOs.

The correlation biplot from Figure 5A has two principal components which explain the total data variances (PC1 = 62.52% and PC2 = 37.48%). All antibacterial activities and constituents are linked to PC1, while only four compounds are associated with PC2 (Linalool, Eugenol acetate, Eugenol, and Caryophyllene). The correlation matrix from the Appendix A and Figure 5A indicates that Eucalyptol, Limonene, o-Cymene, γ-Terpinene, and Camphor are highly correlated with PaIZD and EcIZD, as evaluated by the cylinder technique (r = 0.936–0.976, *p* > 0.05), and strongly correlated with PaIZD1 (r = 0.864–0.918, *p* > 0.05). Limonene and α-Pinen display a good correlation with PaIZD2 and EcIZD2 (r = 0.807–0.873, *p* > 0.05) and a moderate one with PaIZD1 (r = 0.698, r = 0.766, *p* > 0.05). α-Pinen shows a significant negative correlation with SaIZD1 (r = −0.999, *p* < 0.05), and all the others, as previously mentioned, display a high one (r = −[0.961–0.994], *p* < 0.05). EEO contains these phytochemicals, and the PCA analysis confirmed its significant inhibitory activity against all three bacterial species.

Contrariwise, Caryophyllene, Eugenol, and Eugenol-acetate are highly and moderately correlated with SaIZD1 (r = 0.858, r = 0.721, *p* < 0.05). The CEO position in Figure 5A confirms the results from Table 2, Table 3 and Table 4. Neoisomenthol, Isomenthol, Menthofuran, and p-Thymol substantially negatively correlate with SaIZD2 (r = −0.999, *p* < 0.05), while Linalool shows a highly negative correlation (r = 0.990, *p* > 0.05). All exhibit a considerable negative correlation with EcIZD1 (r = −[0.968, 0.994], *p* > 0.05) and a good to moderate one with EcIZD2 (r = −[0.803, 0.711], *p* > 0.05). Therefore, the PEO moderate antibacterial activity on *S. aureus* and the low activity on *E. coli*, as assessed by CT, were confirmed. PEO’s place in the correlation biplot supports its highly inhibitory effect against *S. aureus* evaluated by DDM and confirms its absence against *P. aeruginosa* (Figure 5A).

Figure 5B displays the correlations between the antibacterial activity and phytochemicals quantified in CEO, OEO, LEO, and EEO. In the correlation biplot, the two principal components explain 76.58% of the total data variance (PC1 = 42.81% and PC2 = 33.69%). The antibacterial activities against *S. aureus* and *E. coli* and most phytochemicals are linked with PC1. In contrast, the inhibitory activities on *P. aeruginosa* and Eucalyptol, Limonene, Caryophyllene, and γ-Terpinene are associated with PC2. Eucalyptol and Limonene are significantly correlated with PaIZD (1 and 2). α-Pinen, o-Cymene, and p-Thymol show a good to moderate correlation with EcIZD (1 and 2), r = 0.821–0.589, *p* < 0.05. Finally, the antibacterial activity against *S. aureus* is moderately correlated with p-Thymol, Camphor, Linalool, Linalyl acetate, and Nerol. The EOs’ places confirm that CEO is significantly different regarding its phytochemical profile and antibacterial activity compared to EEO, OEO, and LEO (Figure 5B).

In the correlation biplot from Figure 5C, the two principal components explain 64.20% of the total data variance, with 38.62% attributed to the first (PC1) and 25.58% to the second (PC2). PC1 is associated with SaIZD, EcIZD, and some EOs’ constituents (α-Pinene, Neoisomenthol, Isomenthol, and Menthofuran). At the same time, PC2 is linked with PaIZD and other phytochemicals (Eucalyptol, Lymonene, γ-Terpinene, Camphor, Linalool, Linalyl-acetate, and Nerol). Figure 5C also shows the place of all EOs reported with their phytochemical profiles and antibacterial activities on Gram-positive and Gram-negative bacteria evaluated through both methods.

All interactions expressed as FICI values through DDM and CT are illustrated in Figure 6. Regarding the conventional antibiotics’ influence on the EO binary combinations, TET and NEO diminished EcFICI and PaFICI. Contrariwise, BAC substantially decreased PaFICI and increased EcFICI (Figure 6).

The influence of the EOs’ and conventional antibiotics’ inhibitory activity (expressed as MIC (µg/mL)) on the double and triple combinations and then on interactions (expressed as FICI value) was analyzed by using PCA. All results are shown in Figure 7 and support the data in Table 2, Table 3 and Table 4 and Figure 6.

Figure 8A–C displays the influence of each conventional antibiotic on the first three EO binary combinations; Figure 8D offers an overview of the second group of four EOs and all conventional antibiotics in binary and triple combinations.

When TET is the selected antibiotic (Figure 8A), EcMIC (1 and 2) is moderately correlated with EcFICIs (r = 0.628, r = 0.567, *p* > 0.05). The same observation is available for PaMIC1–PaFICI1 (r = 0.585, *p* < 0.05), while PaMIC2 shows a very low correlation with PaFICI2 (r = 0.075, *p* > 0.05). SaMICs display a minimal negative correlation with SaFICI (r = −0.034, r = −0.192, *p* < 0.05). TET belongs to three triple combinations with partially synergistic action against *S. aureus* (CT) and *E. coli* (DDM).

Figure 8B shows the NEO influence; thus, SaMICs (1 and 2) have a significant moderate to good correlation with SaFICI (1 and 2): r = 0.759, r = 0.892, *p* < 0.05, while PaMIC1 moderately correlates with PAFICI1 (r = 0.670, *p* < 0.05). EcMICs (1 and 2) show a low to moderate correlation with EcFICI (1 and 2), r = 0.469–0.589, *p* > 0.05 (Figure 8B).

Figure 8C, with triple combinations containing BAC, indicates a significant powerful correlation between EcMICs and EcFICIs (r = 0.913, r = 0.850, *p* < 0.05) and PaMIC1 and PaFICI1 (r = 0.865, *p* < 0.05). Moreover, EcFICI1 significantly intercorrelates with EcFICI2, and PaFICI1 reports a moderate intercorrelation with PaFICI2 (r = 0.915, r = 0.654, *p* < 0.05).

In the case of CEO+OEO+EEO+LEO combined with all three antibiotics,
Figure 8D shows that EcMIC2 is significantly correlated with EcFICI2 (r = 0.905, *p* < 0.05), while EcMIC1 has an appreciable correlation with EcFICI1 (r = 0.754, *p* < 0.05). PaMIC1 and SaMIC2 are moderately correlated with PaFICI1 and SaFICI2 (r = 0.607, r = 0.579, *p* < 0.05). Moreover, EcFICI1 is highly correlated with EcFICI2 (r = 0.890, *p* < 0.05) and SaFICI1 moderately correlates with SaFICI2 (r = 0.669, *p* < 0.05).

All combinations and FICI values’ places are illustrated in the correlation biplot (Figure 7A) and the dendrogram with heat map (Figure 7B), supporting our study results.

## 3. Materials and Methods

### 3.1. Materials and Equipment

Commercially available EOs (OEO, CEO, EEO, PEO, and LEO) were provided by “Laboratoarele Fares Biovital SRL” (Orastie, Romania) [27].

EOs used as GC-MS standards were purchased from established manufacturers: Oregano essential oil was provided by Carl Roth GmbH & Co. KG (Karlsruhe, Germany), Clove and Peppermint essential oils were supplied by Sigma Aldrich Corporation (St. Louis, MO, USA), and Lavender and Eucalyptus essential oils were provided by HWI Group (Ruelzheim, Germany). Certificates of Analysis for proving their purity are available in the Appendix A.

All chemicals and reagents were of analytical grade. Conventional antibiotics (Neomycin sulfate, Bacitracin, and Tetracycline hydrochloride) and Poly (ethylene glycol)-block-poly (propylene glycol)-block-poly (ethylene glycol) (Poloxamer 407) were supplied from Sigma-Aldrich Chemie GmbH (Schnelldorf, Germany). Bacterial strains were purchased from authorized suppliers for the sale of ATCC products: *Staphylococcus aureus* (ATCC 25923), *Escherichia coli* (ATCC 25922), and *Pseudomonas aeruginosa* (ATCC 27853). The bacterial strains were cultured in Mueller–Hinton agar (Thermo Fisher, Dreieich, Germany).

The laboratory equipment consisted of an EnSight multimodal plate reader (PerkinElmer Inc., Waltham, MA, USA), adjustable incubator (Memmert GmbH + Co.KG. Büchenbach. Germany), microplate shaking incubator (Heidolph Instruments GmbH & Co. KG. Schwabach, Germany), laminar flow class II microbiological hood (Jouan SA, Saint-Herblain, Pays de la Loire, France), Evoqua double water still (Evoqua Water Technologies GmbH. Barsbüttel, Germany), an electronic balance (Ohaus Corporation, Parsippany, NJ, USA), and NUNC™ MaxiSorp™ 96-well plates (Electron Microscopy Sciences, Hatfield, PA, USA).

### 3.2. GC-MS Analysis

A GC/MS analysis was performed to identify and quantify the constituents of the essential oils of Cloves, Eucalyptus, Lavender, Peppermint, and Oregano by perceptible elucidation, comparing their mass spectra with reference spectra (NIST Library 2020). We used a Thermo Scientific Focus GC (Norristown, PA, USA) with an AI/AS 3000 autosampler coupled with a DSQ II mass detector and equipped with a TraceGOLD TG-624 column 60 × 0.25 × 1.4 (mm). The injection volume was 1 μL, at a 1.3 mL/min flow and a split ratio 1:50 using helium as a carrier gas. The initial oven temperature was set to 110 °C and maintained for 5 min. After, it was increased to 220 °C at the rate of 2 °C/min and maintained for 15 min. The MS transfer line temperature was kept at 240 °C. The ion source temperature was 230 °C, and the electron impact ionization (EI) was set at 70 eV. The spectra were scrutinized in full scan mode over the 50 to 450 mass range, and all constituents’ retention times (RT) were recorded [58].

### 3.3. In Vitro Evaluation of Antibacterial Activity

An in vitro evaluation of the antibacterial activity of the tested solutions was performed using a semi-quantitative method (diffusimetric antibiogram adapted from Kirby–Bauer [59]). The results—expressed as IZD (mm)—were measured after 24 h of contact between the tested solutions and pathogenic bacteria.

#### 3.3.1. Bacterial Inoculum

The bacteria inoculum was obtained using the direct colony suspension method (CLSI) [60]. Hence, a saline suspension (0.9%) of bacterial colonies from a 24 h agar plate was prepared, adjusting it to the 0.5 McFarland standard, with around 10^8^ CFU/mL (CFU = colony-forming unit) [60].

#### 3.3.2. Sample Solutions

The O/W emulsions were prepared with an EO concentration of 30% *w*/*w*; the emulsifier was Poloxamer 407 5% in water, as previously mentioned [27,61].

Each O/W emulsion was diluted with double-distilled water to assess each EO stock solution’s final concentration (25 mg/mL) [27].

The binary EO combinations were prepared in a 1:1 ratio.

The antibiotic drugs (Neomycin, Bacitracin, and Tetracycline) were dissolved in double-distilled water with a stock solution’s final concentration of 5120 µg/mL.

The triple combinations contained equal parts of each constituent (1:1:1).

#### 3.3.3. Technique

The minimum and maximum diameters of the inhibition zones (IZD) were measured in mm, and the arithmetic mean was calculated using one decimal place. Sterile cellulose discs with Ø 6 mm were impregnated with 10 µL of the sample solutions and placed on the Petri plates. Sterile borosilicate glass cylinders with Ø 6 mm were placed on the Petri plates with culture media and loaded with 100 µL of sample solutions.

Then, the Petri plates were incubated at 37 °C and read after 24 h. The minimum onhibitory concentration (MIC) was calculated from the arithmetic mean of IZD.

The negative controls were non-impregnated sterile cellulose discs or non-loaded glass cylinders.

The indicative method to evaluate the Quantity/Effect correlation (Q/Ef index) specific for antimicrobial agents screening consists of MIC value calculation:
$$Effect\;(MIC)=\frac{Q(µg)}{V(µL)}$$

Q—the quantity (µg) of sample solution applied.V—the volume of the environment in which the tested sample diffused and inhibited microbial multiplication.

V was calculated from the average of the IZD, to which the classic cylinder volume formula was applied:

V (µL) = π R^2^ × G


π = 3.14 (the circle constant);R^2^—the square of the radius from the mean IZD (mm);G—the cylinder generator, respectively, the thickness of the culture medium layer (mm).


V (µL) = (ϕ (mm) × 2^−1^)^2^ × 3.14 × 3 (mm)


The fractional inhibitory concentration index (FICI) was calculated for the double and triple combinations as follows:

FICI_AB_ = (MIC_A in combination_/MIC_A alone_) + (MIC_B in combination_/MIC_B alone_)


FICI_ABC_ = (MIC_A in combination_/MIC_A alone_) + (MIC_B in combination_/MIC_B alone_) + (MICc _in combination_/MICc _alone_)


The FICI values were interpreted as follows: FICI ≤ 0.5—synergism; 0.5 < FICI < 1—partial synergism; FICI = 1—additive effects; 1 < FICI ≤ 4—indifference; and FICI > 4—antagonism [39].

### 3.4. Data Analysis

The analyses were performed in triplicate, and the results are expressed by mean value ± SD. The differences between the combination groups were determined using Box and Whisker Plots from Microsoft 365 Excel^®^ v.2023 (Microsoft Corporation, Redmond, WA, USA) [27]. The statistically significant values are marked in Figure 3 with superscripts [60]. The Kruskal–Wallis samples comparison, Blend–Altman comparisons between two different methods of the same determination, and correlations between variable parameters examined through Principal Component Analysis [62] were performed with XLSTAT 2024.1.0. 1418 by Lumivero (Denver, CO, USA) [63]. Statistical significance was established at *p* < 0.05 [64].

## 4. Conclusions

It is not easy to assess the interactions between essential oils and conventional antibiotics in various combinations and appreciate their influence on antibacterial activity. First, the tested Gram-positive and Gram-negative bacteria are crucial regarding their incidence and multidrug resistance. Then, the provenance and storage of essential oils significantly influence the type and amount of bioactive constituents. Next, the differences regarding the phytochemical profile of the combined EOs, the intrinsic mechanisms of the antibacterial actions of the main constituents, and their behavior in combinations are significant. The antibacterial activity evaluation method is essential to diminish bioactive volatile constituents’ potential losses or degradation. When EOs are combined with conventional antibiotics, their structural properties and mechanisms of action are of substantial importance. The present study could enrich scientific databases, offering complex data about the antibacterial activity of five commercially available Essential Oils (Lavender, Clove, Oregano, Eucalyptus, and Peppermint) alone in 5 binary EO associations and 15 triple combinations with three conventional antibiotics (Tetracycline, Neomycin, and Bacitracin). Synergistic and additive effects were obtained between the EOs with different bioactive phytochemicals and moderate inhibitory activity on the tested bacteria. Our study reported one synergic binary combination (OEO+CEO) against *P. aeruginosa*, another partial synergy (EEO+PEO) against *S. aureus*, and another additive one (CEO+PEO) against *E. coli*. Similar observations regarding these antibacterial effects are available for the conventional antibiotics combined with EOs. Previously mentioned binary combinations led to four partially synergistic triple combinations with antibiotic drugs (TET and NEO) that were revealed against *S. aureus* and *E. coli*, three containing TET. The present study could not establish which diffusimetric technique was better, because these results were recorded through antibiotic antibiogram techniques, and all previously mentioned factors were involved. Further research could investigate the inhibitory effects of these combinations against tested bacteria through other methods. Moreover, the antibacterial effects of other potential double and triple combinations of these five essential oils and conventional antibiotics could be explored to find the most effective ones.

## Figures and Tables

**Figure 1 antibiotics-13-00478-f001:**
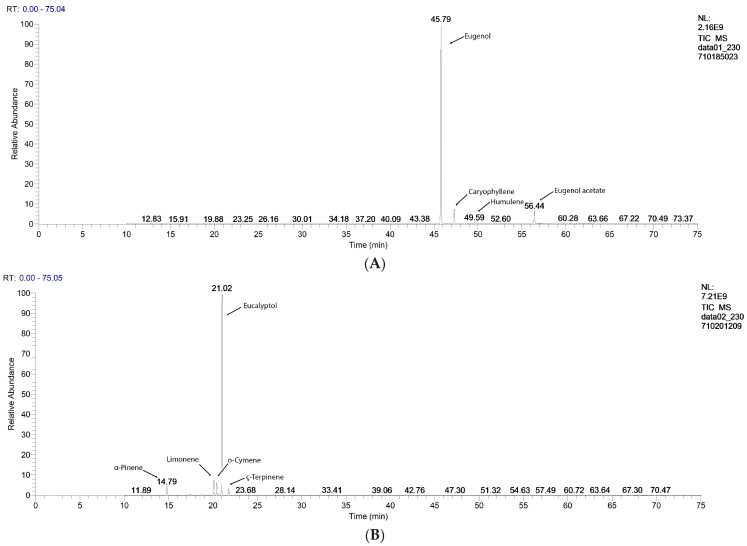
GC-MS chromatograms of EO samples: (**A**). Clove Essential Oil, (**B**). Eucalyptus Essential Oil, (**C**). Lavender Essential Oil, (**D**). Peppermint Essential Oil, and (**E**). Oregano Essential Oil.

**Figure 2 antibiotics-13-00478-f002:**
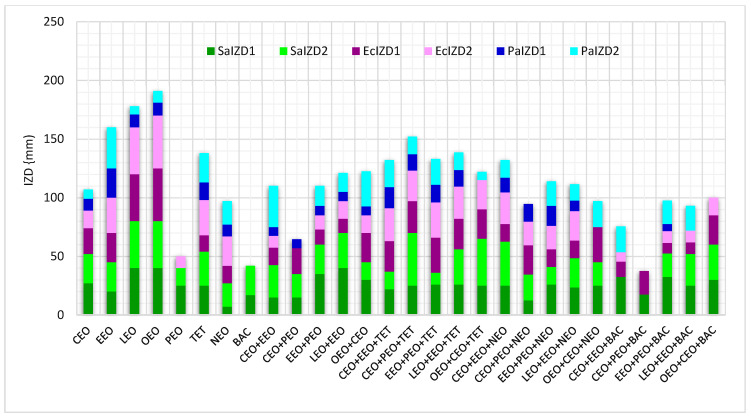
Stacked columns chart, as an overview of antibacterial activity against Gram-positive and Gram-negative bacteria, expressed as IZD (mm). LEO—Lavender Essential Oil; CEO—Clove Essential Oil; OEO—Oregano Essential Oil; EEO—Eucalyptus Essential Oil; PEO—Peppermint Essential Oil; TET—Tetracycline; NEO—Neomycin; BAC—Bacitracin; and IZD—inhibition zone diameter (mm), the scale of measurement was as follows: powerful inhibitory effect at IZD ≥ 35 mm, strong inhibitory effects at 35 > IZD ≥ 25 moderate inhibitory effect at 25 > IZD ≥ 15 mm, mild inhibitory effect when 15 > IZD ≥ 10 mm, and no inhibitory effect at IZD < 10 mm. Sa—*S. aureus*, Ec—*E. coli*, and Pa—*P. aeruginosa*.

**Figure 3 antibiotics-13-00478-f003:**
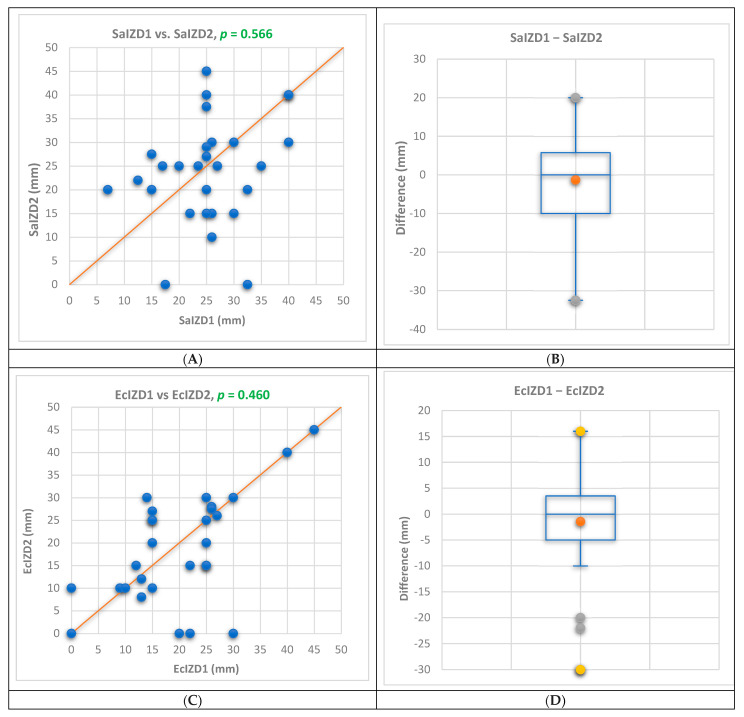
Bland–Altman comparison between both techniques of diffusimetric antibiogram for all bacteria tested: *S. aureus* (**A**,**B**), *E. coli* (**C**,**D**), and *P. aeruginosa* (**E**,**F**). The differences are statistically significant at *p* < 0.05.

**Figure 4 antibiotics-13-00478-f004:**
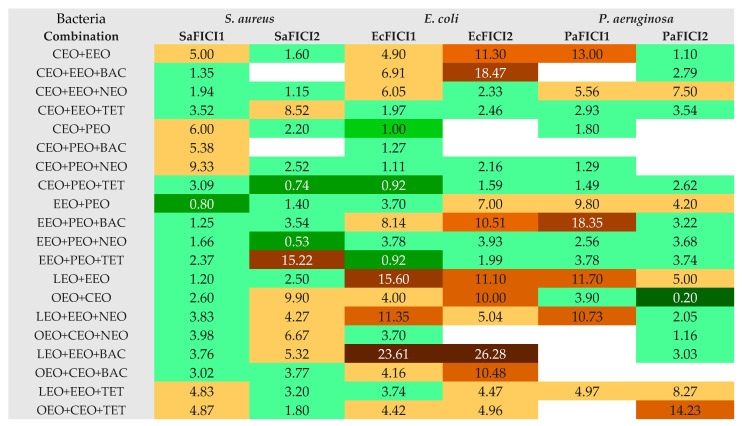
Heat map of FICI values for double and triple combinations. LEO—Lavender Essential Oil; CEO—Clove Essential Oil; OEO—Oregano Essential Oil; EEO—Eucalyptus Essential Oil; PEO—Peppermint Essential Oil; NEO—Neomycin; TET—Tetracycline; and BAC—Bacitracin. FICI—fractional inhibitory concentration index. If IZD = 0 mm, it has no effect; no MIC exists. FICI ≤ 0.5 indicates synergism (S). 0.5 < FICI < 1—partial synergism (PS); FICI = 1—additive effects (Add.); 1 < FICI ≤ 4—indifference (I); FICI > 4—antagonism (Ant.) [39]; 4 < FICI ≤ 10—low antagonism; 10 < FICI ≤ 15—moderate antagonism; 15 <FICI ≤ 20—strong antagonism; FICI > 20—very strong antagonism; FICI1—determined by DDM (disc diffusion method); and FICI2—determined by cylinder diffusion technique. Sa—*S. aureus*, Ec—*E. coli*, and Pa—*P. aeruginosa*. The FICI values ≤ 4 are marked with green and the color intensity decreases from dark green (synergism) to light green (indifference) the FICI values > 4 are marked with brown and the color intensity increases from light brown (low antagonism) to dark brown (very strong antagonism).

**Figure 5 antibiotics-13-00478-f005:**
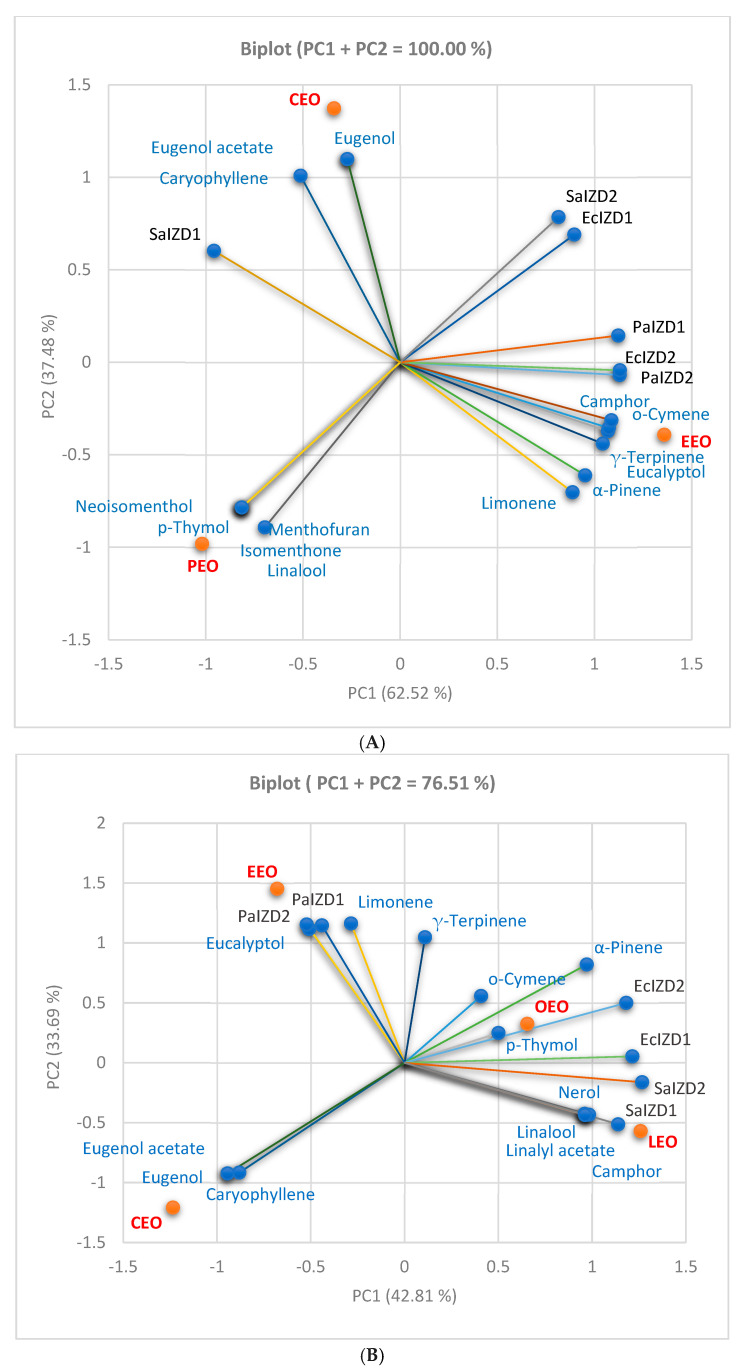
Correlations between EOs’ antibacterial activity (expressed as IZD) and the main phytoconstituents: (**A**) EEO-PEO-CEO group; (**B**) CEO-OEO-EEO-LEO group; and (**C**) all EOs. IZD—inhibition zone diameter (mm), Sa—*S. aureus*, Ec—*E. coli*, Pa—*P. aeruginosa*, LEO—Lavender Essential Oil; CEO—Clove Essential Oil; OEO—Oregano Essential Oil; EEO—Eucalyptus Essential Oil; PEO—Peppermint Essential Oil; IZD1—determined by DDM (disc diffusion method); and IZD2—determined by cylinder technique.

**Figure 6 antibiotics-13-00478-f006:**
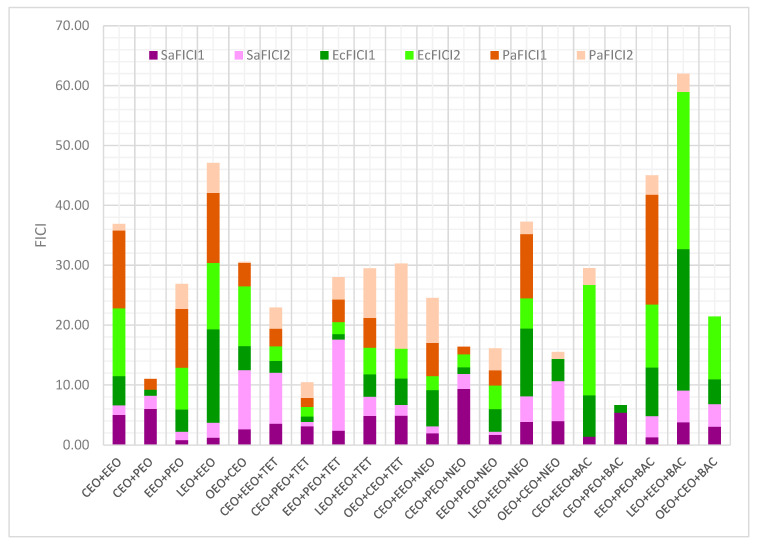
An overview of interactions between EOs and conventional antibiotics against Gram-positive and Gram-negative bacteria, assessed by both techniques of diffusimetric antibiogram and expressed as FICI. LEO—Lavender Essential Oil; CEO—Clove Essential Oil; OEO—Oregano Essential Oil; EEO—Eucalyptus Essential Oil; PEO—Peppermint Essential Oil; NEO—Neomycin; TET—Tetracycline; and BAC—Bacitracin. FICI—fractional inhibitory concentration index. If IZD = 0 mm, it has no effect; no MIC exists. FICI ≤ 0.5 indicates synergism (S). 0.5 < FICI < 1—partial synergism (PS); FICI = 1—additive effects (Add.); 1 < FICI ≤ 4—indifference (I); FICI > 4—antagonism (Ant.) [39]; 4 < FICI ≤ 10—low antagonism; 10 < FICI ≤ 15—moderate antagonism; 15 < FICI ≤ 20—strong antagonism; FICI > 20—very strong antagonism; FICI1—determined by DDM (disc diffusion method); and FICI2—determined by cylinder diffusion technique. Sa—*S. aureus*, Ec—*E. coli*, and Pa—*P. aeruginosa*.

**Figure 7 antibiotics-13-00478-f007:**
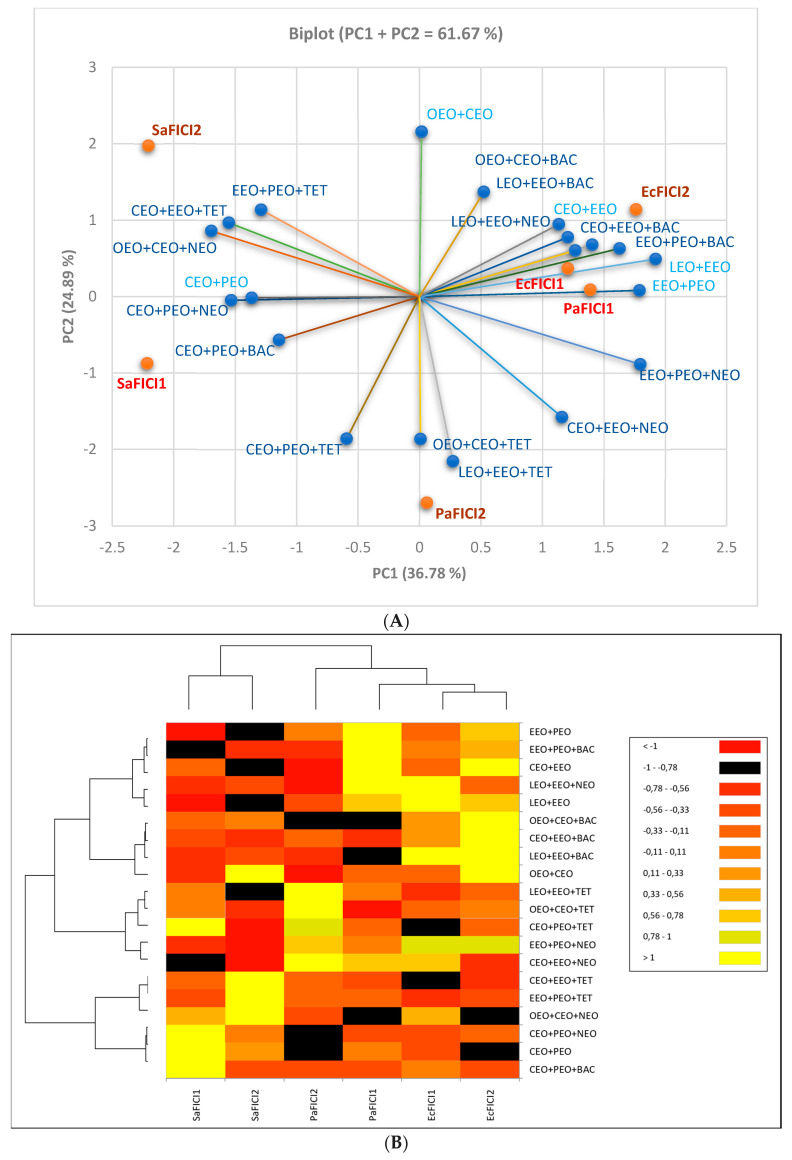
(**A**) Correlations between all tested double and triple combinations and FICI values. (**B**) AHC Dendrogram, with 5 clusters: C1 = LEO+EEO+NEO; C2 = CEO+EEO+NEO; C3 = CEO+EEO+TET; C4 = CEO+PEO; and C5 = LEO+EEO+BAC and heat map. LEO—Lavender Essential Oil; CEO—Clove Essential Oil; OEO—Oregano Essential Oil; EEO—Eucalyptus Essential Oil; PEO—Peppermint Essential Oil; NEO—Neomycin; TET—Tetracycline; and BAC—Bacitracin. FICI—Fractional inhibitory concentration index. FICI1 is determined by DDM (disc diffusion method), and the cylinder diffusion technique determines FICI2. Sa—*S. aureus*, Ec—*E. coli*, and Pa—*P. aeruginosa*.

**Figure 8 antibiotics-13-00478-f008:**
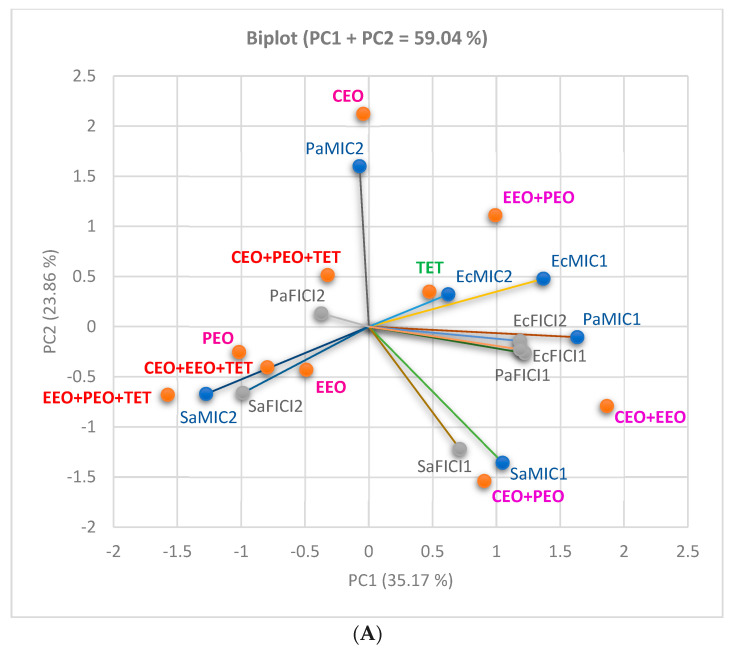
Correlation of MIC values of EOs and those of conventional antibiotics in double and triple combinations and FICI value. (**A**–**C**) CEO+PEO+EEO group in triple combinations with various antibiotics: (**A**) TET, (**B**) NEO, (**C**) BAC, and (**D**) CEO+OEO+EEO+LEO in double and triple combinations, with 3 antibiotics. LEO—Lavender Essential Oil; CEO—Clove Essential Oil; OEO—Oregano Essential Oil; EEO—Eucalyptus Essential Oil; PEO—Peppermint Essential Oil; NEO—Neomycin; TET—Tetracycline; BAC—Bacitracin; MIC—minimum inhibitory concentration (µg/mL), and FICI—fractional inhibitory concentration index. FICI1 was determined by DDM (disc diffusion method), and the cylinder diffusion technique provided FICI2. Sa—*S. aureus*, Ec—*E. coli*, and Pa—*P. aeruginosa*.

**Table 1 antibiotics-13-00478-t001:** The main constituents of EOs quantified by GC-MS analysis expressed as percentages and registered as a mean of 3 determinations ± SD.

RT [min]	Essential Oil	CEO	EEO	LEO	PEO	OEO
	Compound Name	Quantity (%)				
45.79	Eugenol	86.2272 ± 4.8956	-	-	-	-
47.29	Caryophyllene	6.8733 ± 0.4562	-	-	1.6170 ± 0.2035	1.4581 ± 0.2668
56.44	Eugenol acetate	5.7515 ± 0.6057	-	-	-	-
21.00	Eucalyptol	-	83.7478 ± 4.8364	0.7360 ± 0.1087	5.0452 ± 0.2877	0.8904 ± 0.0880
21.77	γ-Terpinene	-	2.2315 ± 0.2701	-	0.2864 ± 0.0166	2.9827 ± 0.3837
31.34	Isomenthone	-	-	-	26.5289 ± 1.5690	-
26.16	Linalool	-	0.0102 ± 0.0019	52.9348 ± 7.4084	0.0670 ± 0.0036	1.4072 ± 0.1055
31.48	Camphor	-	0.0138 ± 0.0009	1.3463 ± 0.2607	-	-
32.23	Neoisomenthol	-	-	-	55.0951 ± 7.7417	-
43.29	p-Thymol	-	-	-	0.0896 ± 0.0147	72.0862 ± 9.3764
35.69	Linalyl acetate	-	-	32.3146 ± 2.5124	-	0.0152 ± 0.0021
42.95	Nerol	-	-	2.1132 ± 0.1975	-	-
20.10	Limonene	-	5.4506 ± 1.0435	1.1219 ± 0.1812	2.0857 ± 0.2091	0.6877 ± 0.0737
20.42	o-Cymene	-	4.1323 ± 0.5402	0.2470 ± 0.0253	0.1803 ± 0.0320	16.2692 ± 1.4297
14.79	α-Pinene	-	2.8165 ± 0.2594	3.1612 ± 0.2142	0.8279 ± 0.1647	2.3971 ± 0.3353
17.46	β-Pinene	-	0.2631 ± 0.0227	0.0478 ± 0.0055	0.8802 ± 0.0767	1.3942 ± 0.2326
30.16	Menthofuran	-	-	-	2.4656 ± 0.3486	-

## Data Availability

Data are available in the present manuscript.

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
