# Peer review of "Phytochemical Screening and Antibacterial Activity of Commercially Available Essential Oils Combinations with Conventional Antibiotics against Gram-Positive and Gram-Negative Bacteria"

_antibiotics, 2024, doi:10.3390/antibiotics13060478_

Round 1

Reviewer 1 Report

Comments and Suggestions for Authors

In the manuscript, the authors explored the anti-bacterial effects of combinations of five different commercially available essential oils.

Following are suggestions to strengthen the impact of the manuscript:

1.     Figure 1, please label the main constituents beside their corresponding peak in the GC-MS chromatograms.

2.     Line 128, how to understand “Based on literature data and phytochemical profile, …”? Could the authors explain more about the rationale why choose these 5 binary combination pairs not the other pairs?

3.     Line 149, “35>IZD>=25”, “mm” unit is missing here.

4.     Table 2/3/4, I would suggest to put the defined scales for IZD and FICI in the table followed by their original values in brackets or color code the values based on the defined scales. This will help readers to better compare the quantified effects among different samples.

5.     Some binary combination pairs showed reduced inhibitory activity (IZD) compared to applying them alone. How to explain this?

6.     Figure 2, the way of how the data presented here is a little confusing. Maybe try to combine the four plots into one grouped bar chart? Bars are grouped by position for different measurements (SaIZD1, SaIZD2, EcIZD1, EcIZD2, PaIZD1, PaIZD2) with color indicating different sample combinations (EOs, Standard antibiotics, EOs double combinations, Triple combinations EOs+NEO, Triple combinations EOs+TET, Triple combinations EOs+BAC). P-values can be performed between samples in the same measurement methods.

7.     Figure 4, the authors had explained the correlations between main components and measured activity revealed by PCA charts. Then what is the further conclusion from this analysis? Which measurement method gave a higher confidence when comparing the inhibitory effect of different samples?

8.     Figure 5, by elucidating the correlation between MIC and FICI, what is the conclusion? How this analysis helped in explaining the experimental results?

9.     Figure 6, seems like there is no trend in each cluster. How to explain this?

10.  Figure 7, same as figure 2, please use grouped bar chart to re-plot the data for a better comparison among samples.

Author Response

Dear Reviewer 1, 

Thank you so much for your time and professionalism in reviewing our MS. We are grateful for all your accurate comments and suggestions to organize our MS better, thus significantly increasing its quality. 

We sent you our responses in the attachment.

Reviewer 2 Report

Comments and Suggestions for Authors

General Comments:

Antibiotic resistance poses a significant threat to human health. In a study by Răzvan Neagu et al., the antibacterial properties of five commercially available essential oils were evaluated against well-known multi-drug resistant strains both individually and in combination. It is essential to thoroughly reorganize and revise the manuscript. I am confident that the clarity and coherence of the scientific writing in English would greatly benefit from careful copyediting before publication.

Major Essential Revisions:

1. The manuscript title should be optimized to highlight the author's selection of drug-resistant Gram-positive and Gram-negative bacteria.

2. The manuscript contains several figures showing similar or identical experimental outcomes, necessitating their consolidation and improved presentation to enhance the publication quality.

3. It is imperative to incorporate discussions into each section's results for a comprehensive analysis and interpretation.

4. It is essential to condense the conclusions in the manuscript to emphasize the main idea effectively.

5. While the combination of essential oils and antibiotics demonstrates a positive antibacterial effect, it is crucial to author evaluate the potential toxicity of essential oils and determine the final mode of administration in vivo.

Comments on the Quality of English Language

 It is essential to thoroughly reorganize and revise the manuscript. I am confident that the clarity and coherence of the scientific writing in English would greatly benefit from careful copyediting before publication.

Author Response

Dear Reviewer 2, 

Thank you so much for your time and professionalism in reviewing our MS. We are grateful for all your accurate comments and suggestions to organize our MS better, thus significantly increasing its quality. 

We sent you our responses in the attachment.

Reviewer 3 Report

Comments and Suggestions for Authors

In the present manuscript, Neagu et al. carried out the phytochemical screening and antibacterial activity of five commercially available essential oils Lavender, Clove, Oregano, Eucalyptus, and Peppermint against the Staphylococcus aureus, Escherichia coli and Pseudomonas aeruginosa alone and in various combinations. The manuscript is scientifically sound, written well, and the results are displayed in figures and tables properly. However, I have a few queries.

1. why the authors choose 5 specific essential oils in the present study?

2. Please mention full forms in the abstract for MIC and IZD.

3. Please add some discussion for the results 

4. Add standard deviation for all values in tables.

Author Response

Dear Reviewer 3, 

Thank you so much for your time and professionalism in reviewing our MS. We are grateful for all your accurate comments and suggestions to organize our MS better, thus significantly increasing its quality. 

We sent you our responses in the attachment.

Round 2

Reviewer 1 Report

Comments and Suggestions for Authors

Questions had been addressed. Agree to accept.